# Reward-free Policy Optimization with World Models

## Abstract

As AI capabilities advance, their rapid progress is not keeping pace with the need for safe and value-aligned algorithms, raising concerns about autonomous systems. E.g., maximizing expected return in reinforcement learning can lead to unintended and potentially harmful consequences. This work introduces Reward-free Policy Optimization (RFPO), a method that prioritizes goal-oriented policy learning over reward maximization by eliminating rewards as the agent's learning signal. Our approach learns a world model that simulates backward in time, and then uses it to construct a directed graph for planning, and finally learning a goal-conditioned policy from the graph. The algorithm has two requirements: (1) the goal has to be defined, and (2) the agent needs sufficient world knowledge, enabling it to plan. This method removes the risks associated with reward hacking and discourages unintended behaviors by allowing for human oversight. Additionally, it provides a framework for humans to build transparent and high-level algorithms by using the (low-level) learned policies. We demonstrate the effectiveness of RFPO on maze environments with pixel observations, where the agent successfully reaches arbitrarily selected goals and follows human-designed algorithms. In conclusion, RFPO enables agents to learn policies without rewards and provides a framework for creating high-level behaviors.

**Keywords:** Reward-free, Goal-conditioned, World Models, Planning

## 1 Motivation

In recent years, the rapid advancement of artificial intelligence (AI) capabilities has brought great opportunities for innovation (Schrittwieser et al., 2020; Brown & Sandholm, 2019; Brown et al., 2020; Radford et al., 2021) and the potential to revolutionize various industries (Levine et al., 2018; Andrychowicz et al., 2020; Kalashnikov et al., 2018). However, the deployment of capable and autonomous AI systems has raised significant concerns (Goodall, 2014; Bostrom & Yudkowsky, 2018; Amodei et al., 2016) about safety (Bengio et al., 2023), unintended emerging behavior (Russell, 2019) and misalignment with human values (Christiano et al., 2017; Hadfield-Menell et al., 2016). Currently, many AI systems are trained using Reinforcement Learning (RL) (Ye et al., 2021) or, in the case of large language models (LLMs), through Reinforcement Learning with Human Feedback (RLHF) (Ouyang et al., 2022). While these methods have led to considerable successes, they fundamentally rely on reward functions for behavior optimization. A major concern lies in the fact that a reward function, which is designed to guide the learning process, may not always align with human values. In some cases, the agent may find hidden loopholes to maximize the reward signal in a way that is not intended by its designers (Leike et al., 2017; Baker et al., 2019).

A famous example of such misalignment is Nick Bostrom's paperclip maximizer scenario (Bostrom, 2017) in which a super-intelligent AI system is tasked to maximize paperclip production. It eventually leads to an astronomical number of paper clips, but resulting in catastrophic outcomes to the ecosystem and society. A game, inspired by this idea, can be played online[1] to understand the full scope of this issue. This thought experiment highlights an important problem: rewards usually represent a proxy target for an underlying objective that is hidden and difficult to formalize (Leike et al., 2017). For example, while the fundamental biological objective (Dawkins, 2016; Williams, 2018)

---

[1] https://www.decisionproblem.com/paperclips/

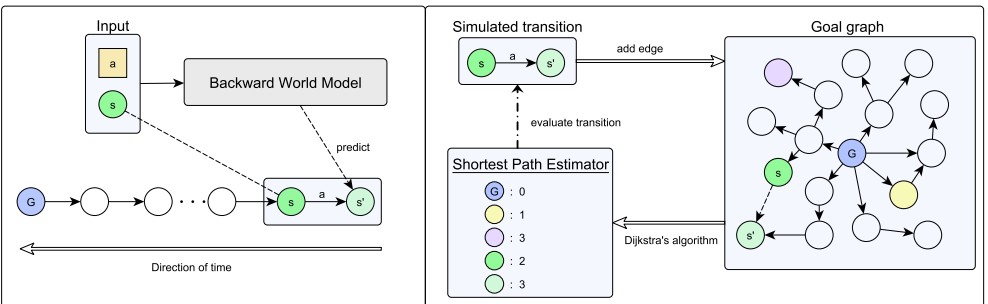

Figure 1: Visualization of the planning process. The left side shows a backward world model that creates simulations from a goal state (blue) and predicts the next state $s'$, but backward in time. The right side visualizes the construction of a goal graph where each simulated transition creates a directed edge. When the graph is completed, the shortest path estimator (SPE) is calculated by estimating the shortest distances to the goal which concludes the planning process. The SPE decides for every simulated transition if it is beneficial for policy optimization or if it should be ignored.

for many organisms is to maximize evolutionary fitness, evolution through natural selection was not able to directly encode this goal into organisms (Mayr, 1982; Dennett, 1995). Instead, from this process emerged reward-bounded proxies like hunger or the need for social interaction which indirectly contribute to the overarching goal of survival and reproduction. Similarly, as AI systems get optimized on abstract objectives (Bach, 2011), relying on rewards as proxy (Baker et al., 2019; Silver et al., 2021) can lead to misalignment, as these might not fully capture or misrepresent the true goal. Therefore, we should try to set up AI systems correctly from the start, as corrections may be impossible later on (Yudkowsky et al., 2008). We realize that rewards also have their downsides, so it is worth the effort to explore ways to learn policies without them, which is the goal of this paper.

We believe, as AI becomes increasingly competent and autonomous, it is essential to develop methods that prioritize goal-oriented learning over raw reward maximization that might lead to unintended behavior. This requires a shift from maximizing expected return to achieving desired state configurations that are aligned with human preferences. On top of that, we need interpretable but also hierarchical agent behavior that can be created and modified by humans. Although RFPO won't solve all problems entirely, we strongly believe that it provides new ideas to enable more transparent and controllable AI systems and hope it will inspire further research in this area.

**Our contributions:**

1. **Reward-free Learning:** RFPO learns complex behaviors without using rewards for optimization. Neither a value or an action value function is required, nor is it necessary to calculate the return.

2. **Graph-Based Planning:** Our approach learns a world model to generate simulated data and directed graphs which allow to plan and thereby collect optimal transitions for policy optimization.

3. **Multi-Goal Conditioning:** RFPO introduces the capability to condition a policy on either a single goal or multiple goals simultaneously.

4. **Human-Centric Behavior Framework:** Our approach offers a programming framework that enables humans to create and modify high-level behavioral algorithms by leveraging the (low level) learned policies.

We mention in passing that our algorithm includes characteristics of Kahnemann's System 1 and 2 (Kahneman, 2011): the learned goal policy corresponds to intuitive and the planning (and human framework) to analytical thought processes (System 1 and 2).

## 2 METHOD

We consider reward-free Markov decision processes (MDP) with discrete time steps $t \in \mathbb{N}$, states $s_t \in \mathcal{S}$, and discrete actions $a_t \in \mathcal{A} = \{1, \ldots, m\}$, which come from a policy $a_t \sim \pi(a_t \mid s_t)$. Episodes terminate based on a boolean variable $d_t \in \{0, 1\}$ and the transitions of states and termination indicators follow the environment dynamics $(s_{t+1}, d_t) \sim p(s_{t+1}, d_t \mid s_t, a_t)$. Finally, the objective on the reward-free MDP is to derive a policy $\pi$ that maximizes the probability of reaching one of the defined goal states $s^* \in \mathcal{S}^* \subset \mathcal{S}$. A normal MDP can be viewed as a reward-free MDP by ignoring the reward function and adding a set $\mathcal{S}^*$ of goal states.

In the following, we introduce Reward-free Policy Optimization (RFPO) which is a goal-driven approach that eliminates rewards as the learning signal. This method is based on the idea to plan backwards in time from goal states to as many previous states as possible. To facilitate the planning we discretize the observed states $s_t$.

### 2.1 MODEL ARCHITECTURE

**Discretization.** A key component is the discretization of the state space by using discrete outputs for the representation and world model. In the continuous setting, even small prediction errors from the world model can lead to a fuzzy projection area around the correct target state. Such errors make it challenging to construct accurate graphs for planning, as for each simulation step we face the problem of deciding to which state the prediction $\hat{s}_t$ belongs to. Formally, it can be described as

$$s_t = \arg\min_{s \in \mathcal{S}} \|s - \hat{s}_t\|, \tag{1}$$

where $\mathcal{S}$ is the set of all possible states. Note, that this formulation might not be feasible or might not represent a good solution as the variance of predition errors can differ locally. We avoid these issues by discretizing the state space $\mathcal{S}$ into distinct latent codes $\mathcal{Z}$, hereby ensuring convergence to valid representations. The discretization leads to partially overlapping simulation trajectories, that enable us to construct robust graphs for planning. The discrete representation is learned with an encoder that performs this mapping, which is described in the following section.

**Representation model.** In our theory, the encoder is an bijective function that maps the set of states $\mathcal{S} = \{s_1, s_2, \ldots, s_n\}$ to a latent set $\mathcal{Z} = \{z_1, z_2, \ldots, z_n\}$. We implement this representation model as an Encoder-Decoder architecture (Ballard, 1987) that uses the categorical latent representation of DreamerV2/V3 (Hafner et al., 2023; 2020) with so-called straight-through gradients. Specifically, we represent it probabilistically as

$$z_t \sim q_\theta(z_t \mid s_t), \tag{2}$$

where $\theta$ is the parameter vector. In this formulation each state $s_t$ induces a probability distribution over the latent states. In order to get a deterministic (i.e., approximately bijective) mapping, we additionally minimize the entropy $H(q_\theta(z_t|s_t))$ towards the end of the optimization.

The second component is a decoder that deterministically maps elements of the latent space back to the states, which we write probabilistically as,

$$s_t \sim p_\theta(s_t \mid z_t), \tag{3}$$

even though the mapping is deterministic. Following DreamerV2, the mapping is a transposed CNN. The vector $\theta$ collects the parameters of the encoder and decoder.

The representation loss is

$$\mathcal{L}^{\text{Repr}}(\theta) = \mathbb{E}_{q_\theta(z_t|s_t)} \Bigg[ \underbrace{-\log p_\theta(s_t|z_t)}_{\text{reconstruction loss}} + \beta \cdot \underbrace{\max\big(H(\,q_\theta(z_t|s_t)), \tau\big)}_{\text{entropy loss}} \Bigg] \tag{4}$$

and consists of two objectives: The first component describes the classic reconstruction loss to minimize the decoder's image reconstruction error using a binary cross entropy loss. The second part adds the entropy loss term with a threshold $\tau = 0.05$ and loss weight $\beta$ that increases linearly over time. The schedule of $\beta$ is based on the current training epoch counter $e_c$ and terminal training epoch $e_T$ as follows

$$\beta = \begin{cases} 0 & \text{if } e_c < 0.9 \cdot e_T, \\ \frac{e_c - 0.9 \cdot e_T}{0.1 \cdot e_T} \cdot 5 \times 10^{-6} & \text{otherwise,} \end{cases} \tag{5}$$

such that the entropy $H(q_\theta(z_t|s_t))$ decreases to a minimum in the end. This ensures that the encoder becomes bijective since each state should be represented by one latent $z_t$ in order to build robust graphs without facing node ambiguity for each state. For the latent states, we build a backward world model.

**Backward world models.** In principle, a backward world model could be directly defined for the observed states $s_t$. However, for the subsequent graph construction, discrete latent representations are necessary. For that reason we define the backward world model on the level of the latent states $z_t$. Since we go backward in time, we predict the latent state $z_t$ from the *next* latent state $z_{t+1}$ and action $a_t$. Formally, we write it as a probability distribution

$$z_t \sim p_\theta(z_t \mid z_{t+1}, a_t), \tag{6}$$

where $p_\theta(z_t \mid z_{t+1}, a_t)$ is implemented as a neural network and $\theta$ is the parameter vector. The network employs a Multi-Layer Perceptron (MLP) (Rosenblatt, 1958). Each of the three layers has 2048 units and uses SiLU non-linearity (Elfwing et al., 2018) with layer normalization (Ba et al., 2016). By using a backward world model, we reverse the conventional forward predictions in model-based reinforcement learning (MBRL) (Robine et al., 2023; Ha & Schmidhuber, 2018; Hafner et al., 2019; Moerland et al., 2023; Hafner et al., 2023) to ensure the generation of trajectories that always reach a specified goal. The latent representation of goal states $s^*$ is written as $z^*$. By simulating backwards from $z^*$, RFPO synthesizes trajectories $[z^*, a_t, z_t, a_{t-1}, z_{t-1}, \ldots, a_{t-k}, z_{t-k}]$ that are reversed to obtain goal-reaching forward trajectories $[z_{t-k}, a_{t-k}, \ldots, z_{t-1}, a_{t-1}, z_t, a_t, z^*]$.

The world model loss minimizes the discrete Kullback–Leibler divergence between the encoder and backward world model distribution,

$$\mathcal{L}^{\text{WM}}(\theta) = \mathbb{E}_t\Big[\text{D}_{\text{KL}}\big(q_\theta(z_t|s_t) \,\|\, p_\theta(z_t|z_{t+1}, a_t)\big)\Big]. \tag{7}$$

The expectation sub-indexed by $t$ means that we take the expectation over tuples $(s_t, a_t, s_{t+1})$ sampled from the environment and then sample $z_{t+1} \sim q_\theta(z_{t+1}|s_{t+1})$.

Both models are trained jointly to enable gradient flow back into the encoder parameters, allowing the world model to influence the encoding structure in the latent space. The overall loss function includes both models and is

$$\mathcal{L}(\theta) = \mathcal{L}^{\text{Repr}}(\theta) + w_{\text{wm}} \cdot \mathcal{L}^{\text{WM}}(\theta), \tag{8}$$

where the loss weight for the world model is set to $w_{\text{wm}} = 0.0025$ throughout all experiments.

## 2.2 Graph construction and planning

Given a backward world model, we next describe how to construct a graph for planning (also see Figure 1), that helps later to select good transitions for policy learning. Since the whole planning process takes place in the latent space, we will call the latent states in the following simply states. Algorithm 1 illustrates the planning process: initially, we generate numerous simulations from a given goal state $z^*$ using the backward world model. These simulations serve as the foundation for constructing a graph and then for identifying sub-goals to start further exploration. By continuously generating new simulations based on these sub-goals and iteratively updating the graph, the algorithm incrementally creates a map of the (latent) state space. Eventually, the planning results in a shortest path estimator called SPE. More details for each step follow in the next paragraphs.

**Simulations.** Each simulation starts at a goal state $z^*$ (or at a selected sub-goal) and unrolls it into the past to create trajectories $[z^*, a_t, z_t, a_{t-1}, z_{t-1}, \ldots, a_{t-k}, z_{t-k}]$ of the environment. These simulations use random actions to avoid any selection bias. After completion, each sequence is split into single-step transitions $(z^*, a_t, z_t), (z_t, a_{t-1}, z_{t-1}), \ldots, (z_{t-k+1}, a_{t-k}, z_{t-k})$.

**Graph.** The single-step transitions create a directed graph $G = (V, E)$. Every node $n_{z_t} \in V$ represents a state $z_t$ and each edge $n_{z_t} \to n_{z_{t-1}}$ represents a transition, but *backward* in time. For graph construction, we consider each single-step transition $(z_t, a_{t-1}, z_{t-1})$ and add nodes $n_{z_t}, n_{z_{t-1}}$ and edge $n_{z_t} \to n_{z_{t-1}}$ if they do not yet exist. The actions are not assigned to an edge or added to the graph. They will be used later in the policy learning process. Furthermore, each node $n_{z_t}$ has a visit counter $v(n_{z_t}) \in \mathbb{N}$, which is incremented with each occurrence of $z_t$ in the simulations.

**Sub-goal selection.** The expansion of the graph by sub-goal simulations is an important component. This process allows RFPO to iteratively explore the state space and extend the goal reachability as far

---

**Algorithm 1** World Model Planning Algorithm

---

1: Given: goal $z^*$
2: **procedure** PLAN($z^*$)
3:     $\mathcal{D} \leftarrow$ SIMULATEBACKWARDS($z^*$)                ▷ Generate initial simulations
4:     $G \leftarrow$ CONSTRUCTGRAPH($\mathcal{D}$)                ▷ Build graph from simulations
5:     **while** computing time available **do**
6:         $\mathcal{S} \leftarrow$ SELECTSUBGOALS($G$)                ▷ Select sub-goals
7:         $\mathcal{D} \leftarrow$ SIMULATEBACKWARDS($\mathcal{S}$)                ▷ Generate new simulations
8:         $G \leftarrow$ UPDATEGRAPH($G, \mathcal{D}$)                ▷ Expand graph with new simulations
9:     **end while**
10:     SPE $\leftarrow$ DIJKSTRA($G, z^*$)                ▷ Calculate shortest path estimator (SPE) for $z^*$
11:     **return** SPE
12: **end procedure**

---

as possible. The selection mechanism is inspired by the inverse visit frequency, following the return strategy from the exploration-based algorithm Go-Explore (Ecoffet et al., 2021): we calculate the weight for each node as

$$w_{z_t} = \begin{cases} \frac{1}{\sqrt{v(n_{z_t})+1}} & \text{if } v(n_{z_t}) \geq \delta, \\ 0 & \text{otherwise.} \end{cases} \tag{9}$$

Here, $v(n_{z_t})$ is the visit counter, and $\delta = 15$ is a threshold for the minimum number of visits required to be considered a sub-goal candidate. This weighting ensures that nodes with fewer visits are more likely to be selected. The sub-goals are randomly sampled proportional to these weights. Once sub-goals are selected, backward simulations are started again to update the graph and its visit statistics.

**Shortest path estimator (SPE).** Given the directed graph, we apply Dijkstra's shortest path algorithm (Dijkstra, 1959) to determine the shortest paths from all nodes to the goal, resulting in the shortest path estimator

$$\text{SPE}(n_{z_t}) := \text{"shortest distance from node } n_{z_t} \text{ to goal } n_{z^*}\text{"}. \tag{10}$$

For the goal node itself, we get $\text{SPE}(n_{z^*}) = 0$.

Later on, the SPE acts as a filter on the simulations to remove bad artifacts, such as loops or inefficient action selection. These naturally occur due to non-optimal policies, such as the random policy, that is used for the backward simulations. The SPE represents the result of our planning effort and will be used in Section 2.3 to guide the policy optimization.

**Conditioning on multiple goals**. It is feasible to simultaneously plan and condition a policy on multiple goals. This process begins with backward simulations from each intended goal $[z_1^*, \ldots, z_m^*]$, which are then integrated into a single graph. When applying Dijkstra's algorithm for the SPE, all goal nodes are considered as starting points, with their distances set to zero. This ensures that a policy considers all goals as its targets and acts according to goal reachability. The SPE will guide the policy to reach the closest goal. In this context, note that these directed graphs do not need to be weakly connected or have any other special property.

## 2.3 POLICY OPTIMIZATION

In principle, the raw forward trajectories $[z_{t-k}, a_{t-k}, \ldots, z_{t-1}, a_{t-1}, z_t, a_t, z^*]$ generated by the backward world model seem to be ideal for training a policy via imitation learning, because each sequence reaches the goal state. However, these simulations cannot be carelessly used for optimization due to potential problems such as loops and sub-optimal action selection. This issue has been extensively investigated in the literature (Wu et al., 2019; Wang et al., 2021b), highlighting the need for careful data curation to exclude problematic state-action pairs. By leveraging the shortest path estimator (SPE) derived from the graph, we can evaluate the world model simulations and decide whether a generated state-action pair $(z_t, a_t)$ should be used for policy learning or not. This process essentially transforms randomly generated simulations into useful artificial expert demonstrations to learn from. For each pair $(z_t, a_t)$, we look at its forward transition $(z_t, a_t, z_{t+1})$ and check with the shortest path estimator whether $z_{t+1}$ gets the agent closer to the goal. If so, the transition is added to

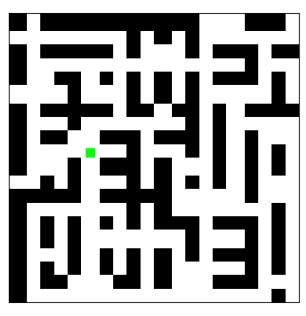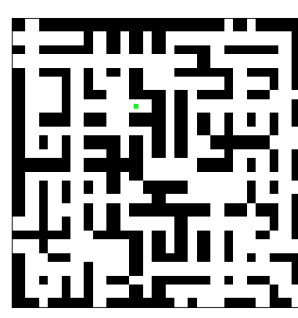

(a) Maze (15×15) observation.    (b) Maze (20×20) observation.    (c) Maze (25×25) observation.

Figure 2: Maze environment: Panel (a) shows an observation with four overlayed goal positions in each corner (marked as orange, cyan, yellow, violet). Panel (b) and (c) show the raw observation for different maze sizes with agent position in light-green.

the policy dataset $\mathcal{D}$ which is used to train the policy. Formally, this selection can be described as

$$\text{SPE}(n_{z_{t+1}}) < \text{SPE}(n_{z_t}) \Rightarrow (z_t, a_t) \in \mathcal{D}. \tag{11}$$

Note, that this dataset has no loops and contains only paths that directly lead to the goal according to the world model. In fact, it uses the best possible path that was found by integrating the results over all simulations.

Finally, we learn a policy $\pi_\nu(a_t|z_t)$ using the classic imitation learning objective, that maximizes the log-probability for the selected state-action pairs $(z_t, a_t)$:

$$\mathcal{L}^{\text{policy}}(\nu) = \mathbb{E}_t \left[ -\log(\pi_\nu(a_t|z_t)) + c_1 S[\pi_\nu](z_t) \right] \ \text{ with } \ (z_t, a_t) \in \mathcal{D}. \tag{12}$$

We add a small action space entropy bonus $S$ with weight $c_1$ following PPO (Schulman et al., 2017). The entropy bonus is helpful if $\mathcal{D}$ contains sub-optimal samples. Again, the expectation sub-indexed by $t$ means that we take the expectation over tuples $(z_t, a_t)$ sampled from $\mathcal{D}$.

### 2.4 HIGH-LEVEL BEHAVIOR ALGORITHMS BY HUMAN DESIGN

In this paper, we draw inspiration from the concepts of Marvin Minsky's "Society of Minds" (Minsky, 1988), which suggests that intelligent behavior arises from the interactions among numerous simpler processes, similar to how a society functions through the contributions of its individuals. By leveraging the SPE as a critic (as defined in Society of Minds), our method allows human or human-aligned LLM involvement (Ahn et al., 2024) for goal formulation and more importantly for programming of high-level behaviors using the learned low-level policies. In our case, this translates to multiple policies (or "minds") working collaboratively yet independently. For instance, a recovery policy can be activated when the currently active policy fails, which is evaluated by the critic. In the experimental section we develop some high-level behavior algorithms and apply them.

**Automatic progress tracking**. Each learned policy memorizes its assigned goal and can continuously evaluate progress towards this target through the shortest path estimator, which should be stored along the corresponding policy. This allows high-level algorithms to monitor the distance to the goal and possibly switch policies once a goal is reached or no reasonable progress is made.

### 3 EXPERIMENTS

#### 3.1 QUANTITATIVE EVALUATION ON MAZE ENVIRONMENTS

To evaluate our method, we start by conducting a series of experiments within maze environments of different sizes ($15 \times 15, 20 \times 20, 25 \times 25$). In this environment, the agent has five actions (left, right, up, down, hold) and receives observations as $64 \times 64$ pixel images in RGB with a bird's-eye view of the entire maze (see Fig. 2). This setup is chosen to select goals based on visual inputs and intuitive human oversight. Since this is a reward-free MDP, there is no reward function.

Table 1: Quantitative evaluation on different maze sizes with 10 runs for each result. Each policy is conditioned on one goal, which is a maze corner. Meanwhile for each column, a different number of sub-goal iterations is considered in order to build the graph. Each sub-goal iteration selects 250 sub-goals from the graph and creates simulations with 100 timesteps each. By increasing the number of iterations, the policies are able to return to their goal from more distant locations.

| Maze size | Goal | Return positions (with different sub-goal iterations) | | | |
|---|---|---|---|---|---|
| | | Iterations: 0 | Iterations: 1 | Iterations: 3 | Iterations: 5 |
| 15 × 15 Fields: 127 | Orange | 89.4 (70%) | 119.7 (94%) | 123.0 (97%) | 124.9 (98%) |
| | Cyan | 80.6 (63%) | 114.0 (90%) | 124.7 (98%) | 124.7 (98%) |
| | Yellow | 53.5 (42%) | 114.6 (90%) | 124.3 (98%) | 126.3 (99%) |
| | Violet | 41.7 (33%) | 89.6 (70%) | 125.2 (98%) | 125.1 (98%) |
| 20 × 20 Fields: 235 | Orange | 77.4 (33%) | 133.0 (56%) | 205.5 (87%) | 223.5 (95%) |
| | Cyan | 74.3 (32%) | 147.9 (63%) | 219.8 (93%) | 227.6 (97%) |
| | Yellow | 81.8 (35%) | 141.7 (60%) | 217.7 (93%) | 228.2 (97%) |
| | Violet | 97.7 (42%) | 163.6 (70%) | 211.4 (90%) | 231.1 (98%) |
| 25 × 25 Fields: 350 | Orange | 80.2 (23%) | 161.2 (46%) | 274.6 (78%) | 326.5 (93%) |
| | Cyan | 95.9 (27%) | 172.0 (49%) | 257.2 (73%) | 306.4 (87%) |
| | Yellow | 52.3 (15%) | 110.7 (32%) | 181.5 (52%) | 252.9 (72%) |
| | Violet | 63.1 (18%) | 123.0 (35%) | 234.7 (67%) | 299.7 (86%) |

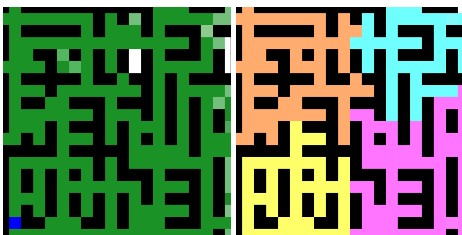

| Maze size | Return positions | Went to closest goal |
|---|---|---|
| 15 × 15 | 124.7 (98%) | 124.7 (100%) |
| 20 × 20 | 221.9 (95%) | 219.9 (99%) |
| 25 × 25 | 335.6 (96%) | 334.5 (99%) |

Table 2: From how many locations do we reach the goal (middle column) or even reach the closest goal (right column)?

Figure 3: Reachability (left) for single goal (only white spots do not reach the goal), and (right) for multi-goal policy.

In this experiment, we selected the four corners of each maze as goal locations, illustrated for the 15 × 15 maze in Figure 2a. To test our approach we trained five policies: four single-goal policies for each corner and one four-goal policy that has all four corners as its goal. Note that integrating multiple goals into a single policy involves minimal modifications to the graph which are described in Section 2.2.

We start by training the backward world model on random trajectories. Depending on the chosen goal set, the graph and the SPE is computed. After that the corresponding policy is trained using imitation learning. For testing a policy, we place the agent at all possible locations throughout the maze. Each position is tested three times to see if the policy could consistently reaches its designated goal. To ensure that successes are not happening by chance, we require for success that the goal is reached within 1.5 times the shortest possible distance to the goal. The left panel of Figure 3 visualizes the success probabilities for a single-goal policy (goal is bottom left): the green locations in the maze all reached the goal fast enough. The light green spots did not always reach the goal in time. From the white locations the agents was not able to reach the goal. The colors in the right panel of Figure 3 show which goal the four-goal policy successfully goes to.

The quantitative results are shown in Table 1 for the single-goal policies and in Table 2 for the four-goal policy. Both experiments validate that the learned policies reach their goals from almost any position within the maze, even from distances as far as 40 steps in the case of our large 25 × 25 maze. These experiments demonstrate that, even in the absence of rewards, it is possible to effectively learn complex policies.

| Method | BlockBadBehaviorEnv | | |
|---|---|---|---|
| | No wall | Barrier 1 | Barrier 1 & 2 |
| PPO (Schulman et al., 2017) | 100% | 0% | 0% |
| DQN(Mnih et al., 2013) | 100% | 0% | 0% |
| Goal policy (Table 1) | 100% | 0% | 0% |
| HL-RND | 100% | 53% | 8% |
| HL-ADV | 100% | 100% | 100% |

Table 3: Goal reachability of different methods. All methods are trained without barriers, but during evaluation they are activated. Human-designed high-level algorithms can detect with their SPEs if they get stuck and adapt their behavior. Model-free algorithms (PPO & DQN) receive a reward of +1 when the goal is reached, otherwise zero.

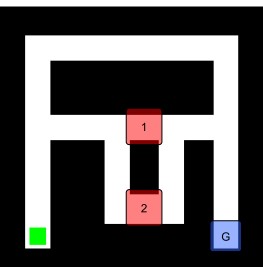

Figure 4: BlockBadBehaviorEnv: start observation with invisible barriers overlayed in red and invisible goal in blue.

For training, the trajectories are collected by acting randomly in the environment. The complete algorithm takes about 30 minutes on one A100 GPU, including the steps for planning and policy training.

### 3.2 HIGH-LEVEL ALGORITHMS BY HUMAN DESIGN

To demonstrate how RFPO helps humans to transparently control AI systems, we consider two scenarios: (i) the BlockBadBehaviorEnv environment, and (ii) SimonEnv environment. In general, programming agents resembles the job of a conductor in an orchestra: While the low-level policies (learned by RFPO) act on their goals, the high-level human-made algorithm supervises them and intervenes when a correction is needed. Hereby, we are able to create novel behaviors that are currently difficult to achieve in the reward-based reinforcement learning setting. The complete Python implementation for each algorithm can be found in Appendix A.

**BlockBadBehaviorEnv**. This environment (see Fig. 4) initially allows certain shortcut solutions, which are blocked during evaluation time, e.g., due to the emergence of unintended behavior. A prominent example occurs in OpenAI's "Hide and Seek" publication (Baker et al., 2019) where the unintended emerging behavior is called "box surfing". BlockBadBehaviorEnv simulates a similar situation by placing hidden walls in the environment during testing. In the case of "Hide and Seek", this corresponds to changing the environment physics or developing other prevention mechanisms. We compare two high-level algorithms based on low-level RFPO policies with PPO and DQN (results are shown in Table 3):

- **HL-RND**. This algorithm (**H**igh **L**evel - **R**a**N**do**M**) aims to maintain progress towards a goal by dynamically adjusting its actions. It monitors the estimated distance to the goal via SPE and, if progress stops for any reason, switches to random actions temporarily. After a short period of random actions, the algorithm resumes using the goal policy and monitors the SPE progress again.

- **HL-ADV**. The algorithm (**H**igh **L**evel - **ADV**anced) extends HL-RND by allowing an additional helper policy, that is trained separately using a human-selected goal state. In our case, this policy is conditioned on a state along the top pathway to improve adaptability at reaching the real goal. The idea is that a supervisor or teacher (e.g. human or LLM) is stepping in and informally expressing: "RFPO, if you get stuck using your preferred strategy, you can also follow the longer path to reach your goal.". HL-ADV primarily uses the goal policy to monitor the estimated distance to the goal via SPE. If progress stops, it temporarily switches to random actions. But should the random action intervention repeatedly not result in progress, the helper policy is activated. Once the helper policy reaches its goal ($SPE(n_z) = 0$, a position in the middle of the top pathway), the algorithm switches back to the goal policy and resumes monitoring the SPE progress.

**SimonEnv**. Inspired by the popular game Simon, we created an environment based on the maze from Fig. 2a. The agent is given a list of goals which have to be reached in the correct order to solve the task. The goals are the four maze corners which can appear repeatedly on the list. The task is

challenging because the list can quite long (in the experiments 10, 25, 50) and there is no feedback during the process. The following algorithm perfectly solves the task:

- **HL-Simon**. The HL-Simon algorithm changes slightly the concept of HL-ADV by incorporating four different policies. Its purpose is to follow a sequence of goals, always switching to the correct policy. For the currently active goal, it monitors the estimated distance via SPE. When the estimated distance reaches zero, the algorithm automatically selects the next goal in the sequence and switches to the appropriate policy.

## 4 RELATED WORK

**AI safety**. In the reinforcement learning literature various ideas how to align AI behavior with human values have been proposed: Deep Reinforcement Learning from Human Preferences (Christiano et al., 2017) integrates human feedback into the training loop, enabling the agent to learn human-aligned behaviors. Leike et al. (2018) propose Value Alignment through Reward Modeling (Leike et al., 1811) to design scalable models that align agent behavior with human values via reward modeling, and Learning from Human Preferences in Reinforcement Learning (Ibarz et al., 2018) combines human feedback and demonstrations to refine agent behavior, even in complex environments like Atari games. In a similar spirit, our work also proposes an approach for value alignment. However, it is purely goal-driven without referring to rewards.

**Goal-conditioning**. Reaching goals has been widely used in combination with reinforcement learning: Schaul et al. (2015) pioneered Universal Value Function Approximators (UVFAs) (Schaul et al., 2015), which extend traditional value functions to incorporate goals as additional inputs, enabling the agent to generalize across different tasks. Florensa et al. (2018) introduced automatic goal generation for RL agents (Florensa et al., 2018), and Nasiriany et al. (2019) presented goal-conditioned imitation learning combined with planning (Nasiriany et al., 2019). Kidambi et al. (2020) develop MOReL(Kidambi et al., 2020), an offline MBRL approach to learn a pessimistic estimate of the underlying MDP and optimize a policy while minimizing the model bias. Chebotar et al. (2021) learn a goal-conditioned Q-function on offline data where they use sub-sequences and relabeling techniques (Chebotar et al., 2021). Reaching goals is also the central idea of our work, however, we differ in the fact that we base the policy learning purely on reaching predefined goals along shortest paths on backward graphs and not by using any reward signal.

**Backward world models.** Unrolling backwards in time is an idea that has been rarely used in reinforcement learning. There are only a few examples, that use backward world models to improve policy learning, including Goyal et al. (2018); Edwards et al. (2018); Lai et al. (2020); Wang et al. (2021a); Chelu et al. (2020). Most similar to our work is Pan et al. (2022), who combine imitation learning with backward world models (Pan & Lin, 2022) where backward simulations are treated as sub-optimal expert demonstrations to improve the expected return of a forward policy. However, none of these papers are fully reward-free and use the backward models for graph construction and for generating goal-reaching trajectories.

## 5 LIMITATIONS

Reward-free Policy Optimization (RFPO) introduces a method to learn policies without rewards, but it has limitations. First, the algorithm requires that the latent trajectory representations overlap in order to construct robust graph structures. If trajectories do not overlap sufficiently, the resulting graph may be sparse and less useful for planning, degrading policy performance. Second, the goals in RFPO must be explicitly identified by humans beforehand, which limits the system's autonomy and requires human intervention for setting up the initial goals. This could be problematic in environments where goals are dynamic or not easily defined. Lastly, RFPO depends on a well-trained world model to generate accurate backward simulations. Inaccuracies in the world model can lead to faulty graph construction and sub-optimal policy learning.

## 6 CONCLUSION

In this paper, we explore a novel approach for reward-free policy learning which we call Reward-free Policy Optimization (RFPO). Using a backward world model and graph search, RFPO directly conditions policies on desired states, eliminating rewards as the optimization signal. This approach can help alleviating issues like reward hacking and unintended behaviors, promoting safer autonomous systems. A key component, the shortest path estimator (SPE), refines sub-optimal simulations into high-quality training data, effectively creating artificial expert demonstrations. Our framework enhances human oversight and control, facilitating transparent, high-level algorithms with learned low-level policies. Experiments in maze environments demonstrate that RFPO enables agents to achieve and monitor via SPE multiple goals that can be employed in sophisticated human-designed algorithms. By going backwards in time from human-aligned goals, instead of carelessly searching for rewards, this work aims to provide a safer approach for learning challenging tasks while obeying to intended behaviors.

ACKNOWLEDGMENTS

This research has been funded/supported by ... .

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

## A APPENDIX

```python
class HL_RND():
    def __init__(self, policy, env, time_window=5, decrease_threshold=3, random_action_duration=12):
        """
        Initialize HL_RND class.

        :param policy: Primary policy.
        :param env: Environment.
        :param time_window: Window for tracking distances.
        :param decrease_threshold: Minimum decrease to continue primary policy.
        :param random_action_duration: Duration of random actions.
        """
        self.policy = policy
        self.env = env
        self.time_window = time_window
        self.decrease_threshold = decrease_threshold
        self.random_action_duration = random_action_duration

        self.estimated_distances = []  # Track estimated distances
        self.random_action_counter = 0  # Counter for random actions

    def reset(self):
        """Reset internal state."""
        self.estimated_distances = []  # Clear distance history
        self.random_action_counter = 0  # Reset random action counter

    def act(self, obs):
        """Select action based on observation."""
        if self.random_action_counter > 0:
            self.random_action_counter -= 1
            action = self.random_action()

            if self.random_action_counter == 0:
                self.estimated_distances = []  # Clear distance history after random actions
        else:
            action, estimated_distance = self.policy.act_and_track(obs)
            self.estimated_distances.append(estimated_distance)
```

```python
            if len(self.estimated_distances) > self.time_window:
                self.estimated_distances.pop(0)

                if not self.has_decreased_sufficiently():
                    self.random_action_counter = self.random_action_duration
                    action = self.random_action()

        return action

    def has_decreased_sufficiently(self):
        """Check if distance decreased sufficiently within time window."""
        initial_distance = self.estimated_distances[0]
        current_distance = self.estimated_distances[-1]
        return (initial_distance - current_distance) >= self.decrease_threshold

    def random_action(self):
        """Select random action from environment's action space."""
        return self.env.action_space.sample()
```

Listing 1: HL_RND algorithm

```python
class HL_ADV():
    def __init__(self, primary_policy, secondary_policy, env, time_window=5, decrease_threshold=3,
        random_action_duration=8, max_random_tries=3):
        """
        Initialize HL_ADV with primary and secondary policies and related parameters.

        :param primary_policy: The main policy for the agent.
        :param secondary_policy: The fallback policy.
        :param env: The environment.
        :param time_window: Number of recent distance estimates to track.
        :param decrease_threshold: Required decrease in distance.
        :param random_action_duration: Duration of random actions if progress stalls.
        :param max_random_tries: Max attempts of random actions before switching to the secondary policy.
        """
        self.primary_policy = primary_policy
        self.secondary_policy = secondary_policy
        self.env = env
        self.time_window = time_window
        self.decrease_threshold = decrease_threshold
        self.random_action_duration = random_action_duration
        self.max_random_tries = max_random_tries

        self.estimated_distances = []  # Track distance estimates
        self.random_action_counter = 0  # Random action counter
        self.random_tries = 0  # Random action attempt counter
        self.use_secondary_policy = False  # Flag to switch to secondary policy

    def reset(self):
        """Reset the algorithm's state."""
        self.estimated_distances = []
        self.random_action_counter = 0
        self.random_tries = 0
        self.use_secondary_policy = False

    def act(self, obs):
        """Select an action based on the current observation."""
        if self.use_secondary_policy:
            action, estimated_distance = self.secondary_policy.act_and_track(obs)
            if estimated_distance == 0:
                self.use_secondary_policy = False
                self.reset()  # Reset for a fresh start with the primary policy
        elif self.random_action_counter > 0:
            # Take random actions during random action period
            self.random_action_counter -= 1
            action = self.random_action()

            if self.random_action_counter == 0:
                self.estimated_distances = []
                self.random_tries += 1

                # Switch to secondary policy if max random tries exceeded
                if self.random_tries >= self.max_random_tries:
                    self.use_secondary_policy = True
                    self.random_tries = 0
        else:
            # Use primary policy for action and distance estimate
            action, estimated_distance = self.primary_policy.act_and_track(obs)
            self.estimated_distances.append(estimated_distance)

            if len(self.estimated_distances) > self.time_window:
                self.estimated_distances.pop(0)

                # Enter random action period if progress is insufficient
                if not self.has_decreased_sufficiently():
                    self.random_action_counter = self.random_action_duration
                    action = self.random_action()

        return action

    def has_decreased_sufficiently(self):
```

```
70        """Check if the distance has decreased sufficiently."""
71        initial_distance = self.estimated_distances[0]
72        current_distance = self.estimated_distances[-1]
73        return (initial_distance - current_distance) >= self.decrease_threshold
74
75    def random_action(self):
76        """Return a random action from the environment's action space."""
77        return self.env.action_space.sample()
```

Listing 2: HL_ADV algorithm

```
1  class HL_Simon:
2      def __init__(self, policies, goal_sequence):
3          """
4          Initialize with a list of policies and the environment.
5
6          :param policies: List of policies [policy0, policy1, policy2, policy3]
7          :param goal_sequence: List of goals to reach sequentially
8          """
9          self.policies = policies
10         self.goal_sequence = goal_sequence
11         self.current_goal_index = 0
12         self.current_policy_index = 0
13
14     def reset(self):
15         """
16         Reset the state of the algorithm to start over.
17         """
18         self.current_goal_index = 0
19         self.current_policy_index = 0
20
21     def act(self, obs):
22         """
23         Decide on the action based on the current policy and goal.
24
25         :param obs: The current observation
26         :return: The selected action
27         """
28         current_goal = self.goal_sequence[self.current_goal_index]
29         policy = self.policies[current_goal]
30
31         action, estimated_distance = policy.act_and_track(obs)
32
33         if estimated_distance == 0:
34             self.current_goal_index += 1
35             if self.current_goal_index >= len(self.goal_sequence):
36                 self.current_goal_index = 0  # Loop back to the beginning if needed
37
38             current_goal = self.goal_sequence[self.current_goal_index]
39             self.current_policy_index = current_goal
40
41         return action
```

Listing 3: HL_Simon algorithm

# B    BROADER IMPACT

The development of methods that are not using rewards for optimization offer potentially advancements in AI safety, alignment, and transparency. By enabling clearer decision-making processes and incorporating human oversight, algorithms such as RFPO support a collaborative human-in-the-loop approach that could lead to safer and more ethical AI outcomes. Such an approach is particularly relevant for applications in fields like healthcare and education, where ethical considerations and predictable behavior are crucial components. Ensuring that AI goals are comprehensive and aligned with societal values is essential to avoid unintended consequences.

