# OpenReview forum: "Reward-free Policy Optimization with World Models"
_ICLR.cc/2025/Conference — ICLR 2025 Conference Withdrawn Submission_

### Official Review · Reviewer_ZHBY · 2024-10-28

**Soundness:** 2
**Presentation:** 2
**Contribution:** 1
**Rating:** 3
**Confidence:** 4

**Summary:**

This paper proposes a novel framework, Reward-Free Policy Optimization (RFPO), which learns complex behaviors without relying on rewards, using a graph-based planning approach to generate optimal transitions for policy learning. RFPO incorporates a backward world model and the Shortest Path Estimator (SPE) to guide the agent’s exploration and achieve multi-goal conditioning, allowing policies to reach either single or multiple goals within a 2D maze environment. The method is framed as human-aligned, with additional high-level algorithms for error correction, aiming to improve safety and adaptability in AI-driven navigation tasks.

**Strengths:**

S1. **Combining Backward World Model with Graph-Based Planning**. The graph constructed from the backward simulations serves as a map of the latent state space, and the **SPE** derived from this graph gives a unique approach to planning trajectories towards goals. By continuously generating simulations and iteratively updating the graph, the method incrementally builds a comprehensive map of the state space, which is a novel combination of world model simulation and dynamic graph construction for planning.

S2. **Use of Sub-Goals in Graph Construction**. Identifying sub-goals from the graph and then using these sub-goals to explore further is an innovative way of guiding exploration. This is different from typical approaches where the agent only focuses on reaching the final goal. Here, the system builds a hierarchical structure of goals and sub-goals, continuously refining the graph by exploring these sub-goals. While hierarchical reinforcement learning itself isn’t new, the specific approach of using a graph to identify sub-goals and then using backward simulations to refine the graph over time appears novel.

S3. **Joint Training of Encoder and Backward World Model**. The fact that the backward world model influences the encoder’s latent space during joint training is a novel twist. Typically, world models might operate on a fixed latent space, but here the backward model actively helps shape the latent space by encouraging it to align with backward dynamics, which could lead to better representations for planning.

S4. **Multiple Goals**. Treating multiple goal nodes as starting points, the policy can consider each as a potential target and can be optimized to reach the closest goal based on the SPE. This can be useful for scenarios where it only matters that the agent reaches any goal, rather than a specified one.

S5. **End-to-End Training**: The method is trained in an end-to-end manner, integrating all components seamlessly without the need for separate, staged training. This unified approach can improve overall efficiency and coherence, allowing the agent to learn directly from raw inputs to goal-reaching behavior.

S6. **Well-Written Method Section**: The Method section is the paper is notably clear and well-structured, providing a detailed and understandable presentation of the approach. The visualization of the planning process in Figure 1 further aids comprehension.

**Weaknesses:**

W1. **Unclear Novelty in Claimed Contributions**: The stated contributions lack clarity regarding what aspects are truly novel and what builds on existing techniques. For example, "Reward-Free Learning" and "Graph-Based Planning" are established concepts in RL, yet it is not clear from the contributions how RFPO’s implementation differs from or advances these methods. Similarly, "Multi-Goal Conditioning" and "Human-Centric Behavior Framework" are vague and offer little insight into what RFPO uniquely contributes. This makes it difficult to discern what is genuinely original in the authors' approach and what is simply a reapplication of existing methods, leaving the overall novelty of the work unclear.

W2. **Misleading definitions**. The authors represent the encoder probabilistically as $z_t∼q_θ(z_t∣s_t)$, although formalizing it as a bijective function from $S$ to $Z$. A bijection is, by definition, deterministic: each state corresponds to exactly one latent vector and vice versa. Calling it “bijective” is misleading because the mapping isn't one-to-one in a strict sense if sampling from a distribution. In practice, the straight-through gradients help in learning an approximate bijection, but the probabilistic formulation violates the strict definition. The authors claim that the additional entropy minimization term is sufficient to obtain a deterministic mapping. Minimizing entropy can lead to a more deterministic solution, but it doesn't guarantee strict determinism. The same issue arises with the decoder definition. Stochastic modeling is not deterministic reconstruction. I suggest the authors loosen the definitions.

W3. **Model Approximation Issues.** The backward world model, being an approximation of true environment dynamics, can generate unreachable or invalid states, especially in regions with sparse data. In particular, the predicted $s_{t-1}$ may not correspond to a valid state, or no action $a_t$ could transition the agent from $s_{t-1}$ to $s_t$. In the latent space, this can result in inconsistent predictions where $z_{t-1}$ may not decode to a valid or reachable state $s_{t-1}$. Over long trajectories, small prediction errors can accumulate, leading to significant deviations from real trajectories and resulting in states that the agent cannot actually reach or encounter in the environment.

W4. **Assumption of Full Observability.** The method relies on the assumption that the agent has access to complete and accurate state information at each timestep, which may not hold in many real-world scenarios where environments are only partially observable. In such cases, the agent would only receive incomplete or noisy observations rather than full states. This assumption limits the method's applicability in more complex settings where tracking hidden states or inferring missing information would be necessary, as the approach does not account for belief states or integrate memory mechanisms like recurrent networks to handle partial observability.

W5. **Dependency on SPE**: The method heavily relies on the SPE for evaluating transitions, filtering trajectories, and guiding policy learning. This poses a risk, as any inaccuracies in the SPE—such as errors in estimated distances or path evaluations—could lead to suboptimal policy performance, misjudged transitions, or inefficient exploration. Without a thorough analysis of SPE’s robustness or alternatives if it fails, the method’s reliability and generalization across environments remain questionable.

W6. **Lack of Interpretability**. Since the method uses pixel-based inputs, the authors could enhance interpretability by visualizing the outputs of the backward world model. Visualizing a sequence of frames generated by the model would directly assess whether it is functioning as intended—specifically, ensuring that the maze layout remains unchanged and only the agent's position updates correctly. Without such visualizations, it's difficult to verify if the model is accurately predicting valid transitions or introducing unintended changes to the environment.

W7. **Weak Experimental Analysis**. The experimental setup lacks essential context and comparison, making it difficult to assess the method's effectiveness.
1. **No Established Environments**: The authors evaluate their method in a simple 2D maze environment with limited discrete actions, rather than in established GCRL environments that offer standardized benchmarks and baseline comparisons. Widely-used environments like robotic tasks in Robosuite [1] as adopted in [2, 3], and continuous control tasks like Ant [4, 5], provide meaningful context for evaluating goal-conditioned RL methods. The absence of a well-known benchmark makes it difficult to assess the relevance or broader applicability of their results.
2. **No Baseline Comparisons**: The paper does not implement or provide results for other GCRL methods within their chosen environment. This omission makes it impossible to determine if their method offers any substantive improvement over existing approaches.
3. **No Analysis of Method Components**: The method incorporates numerous components—such as the backward world model, graph construction and updating, planning, sub-goal selection, SPE, and additional loss terms - without providing a breakdown of each component's contribution to RFPO’s performance. With no ablation study or component-wise analysis, it is difficult to understand which parts are essential to the method's success or to diagnose how it functions in practice.
4. **Additional Hyperparameters**. RFPO introduces several extra parameters, including: 1) entropy loss term threshold $\tau$, values 0.9 and 5 * 10^-6 used in the $\beta$ tuning schedule, 3) world model loss weight $w_{wm}$, 4) visit counter $\delta$, 5) entropy bonus weight $c_1$. Tuning these values complicates the method’s application in other contexts. Moreover, there is no hyperparameter sensitivity study, leaving it unclear how robust the method is to changes in these parameters or how they might need adjustment across different settings.

W8. **Flawed Design of High-Level Algorithms**. The High-Level algorithms and the environments in which they are evaluated are highly hand-engineered. If anything, section 3.2 highlights the limitations of the proposed goal-conditioned policy and backward world model. The HL extensions do not enhance the original method; they merely introduce policies that intermittently override the main policy without directly addressing its weaknesses. The use of random actions (HL-RND) or policy-switching (HL-ADV) as "corrections" adds little value. Instead of adapting or aligning the main policy dynamically, these methods just swap to other predefined strategies, which doesn’t substantiate claims of advancing AI Safety or alignment.
1. **BlockBadBehaviorEnv**: The authors' concept of "bad behavior" here is artificial and unconvincing. Labeling one maze path as "bad" and another as "good" lacks clear criteria and does not relate to genuine AI Safety or alignment concerns. Unlike OpenAI’s "box-surfing," which involved creative exploitation of physics, this setup simply tests route adaptability by blocking paths during evaluation—a basic test of rerouting, not emergent, unsafe behavior.
2. **SimonEnv**. Rather than being a unique environment, this is essentially a sequence of four episodes from the original environment, each with a different starting location and goal. The novelty this setup adds is minimal.
3. It is not explicitly mentioned whether HL-RND and HL-ADV are used during training to assist policy learning or only during evaluation. I will assume it is the latter.
4. **HL-RND**. Relying on random actions during evaluation might yield some success in a simple 2D maze with limited discrete actions. However, this approach is unlikely to scale or produce reliable outcomes in more complex environments.
5. **HL-ADV**. This setup does not reflect human supervision or alignment, as the authors claim. Training a separate policy to handle a sub-goal and occasionally substituting it for the main policy involves no direct interaction with or adaptation of the original policy. Supervision would entail some signal or interaction to actively guide the primary policy.
6. The comparisons with pure DQN and PPO lack coherence. One could equally:
    1) sample random actions (like HL-RND) when the DQN/PPO policy fails to receive equally high rewards as obtained during training;
    2) switch to an entirely different DQN/PPO policy (much like HL-ADV) that is trained to navigate to the human-specified midpoint at the top of the environment.
7. The authors claim that the HL algorithms promote safer and more ethical AI outcomes, yet they allow agents to take random actions during evaluation as part of their “human-aligned” approach. There is no control over the number of random actions taken or their potential consequences, which undermines the method’s alignment and safety objectives.
8. **HL-Simon**. Sequentially activating predefined policies for individual tasks is simplistic and hardly qualifies as an “algorithm.” The agent has full knowledge of which policy to use for each goal in the sequence, requiring no intelligent coordination. Additionally, the paper presents no results for HL-Simon. The authors simply state that it perfectly solves the task.
9. All of the HL "algorithms" assume that the SPE is functioning effectively, yet there is no thorough examination of its reliability across more advanced settings.

[1] Zhu, Yuke, et al. "robosuite: A modular simulation framework and benchmark for robot learning." *arXiv preprint arXiv:2009.12293* (2020).

[2] Yang, Rui, et al. "Rethinking goal-conditioned supervised learning and its connection to offline rl." *arXiv preprint arXiv:2202.04478* (2022).

[3] Nair, Ashvin V., et al. "Visual reinforcement learning with imagined goals." *Advances in neural information processing systems* 31 (2018).

[4] Nachum, Ofir, et al. "Data-efficient hierarchical reinforcement learning." *Advances in neural information processing systems* 31 (2018).

[5] Chane-Sane, Elliot, Cordelia Schmid, and Ivan Laptev. "Goal-conditioned reinforcement learning with imagined subgoals." *International conference on machine learning*. PMLR, 2021.

**Questions:**

Q1. If the encoder maps a single state $s_t$ to a latent vector $z_t$, why do the authors formalize the encoder as a bijective function that maps from states $S$ to states $Z$?
Q2. The encoder is described as a "bijective" mapping while representing it probabilistically. Could you clarify how this formulation supports a deterministic one-to-one mapping, particularly with the probabilistic nature of the encoder and decoder?
Q3. Given that the backward world model is only an approximation, how does it handle regions with sparse data, and what measures do you have to prevent the generation of unreachable or invalid states?
Q4. Why was the 2D maze chosen over established GCRL environments, such as Robosuite or AntMaze, which offer standardized evaluation metrics and benchmarks?
Q5. Why didn’t you implement or test against other GCRL methods in your chosen environment? Without baselines, it’s difficult to contextualize RFPO’s performance.
Q6. Given that the high-level algorithms rely on rigid policy switching, how would this approach generalize to more complex tasks? Does it have applications beyond simple maze rerouting?
Q7. The HL-RND algorithm relies on random actions to overcome obstacles. How would this approach scale in larger or continuous spaces where random actions are unlikely to yield good outcomes?
Q8. Can you clarify what aspects of RFPO are truly novel compared to established concepts like reward-free learning and graph-based planning in RL? Specifically, how does RFPO advance these areas beyond existing methods?

### Suggestions
1. Define the MDP as a reward-free goal-conditioned MDP, following conventions to prior literature [2, 3, 4, 5].
2. The authors could strengthen the evaluation by using a physics-based continuous control benchmark, such as:
    1. **Mujoco**: Task an ant agent with moving to specific locations on a 2D plane.
    2. **RoboSuite**: Have the robotic arm push, pick up, or move an object to a designated goal location.
3. Presenting the results in Table 1 as a line plot would be more informative, with the x-axis representing iterations and the y-axis showing either the score or success percentage. This would provide a clearer view of performance trends over time and allow for easier comparison between goals or maze sizes.
4. Include experiments with partial observability, where the agent observes only a limited subsection of its surrounding tiles. This would test the method's robustness.
5. I appreciate the multiple references to social psychology literature throughout the paper, but these seem rather far-fetched and disentangled from the authors’ core work.

In conclusion, this work shows potential, particularly if empirically evaluated in more complex scenarios. However, in its current form, it falls short of ICLR standards due to significant gaps in the experimental evaluation. While the methodological approach is interesting and promising, the lack of rigorous empirical validation undermines the ability to assess its practical utility. As it stands, major revisions are needed to fully demonstrate the effectiveness and applicability of RFPO.

---

### Official Review · Reviewer_TaX8 · 2024-10-29

**Soundness:** 1
**Presentation:** 1
**Contribution:** 1
**Rating:** 1
**Confidence:** 5

**Summary:**

This work proposes a method for developing goal-conditioned policies in RL. Their proposal revolves around using a *backwards* world model which, given a goal state, makes backward predictions to different states in a discretised gridworld. Using Dijkstra's algorithm, a shortest path from different states to the goal is found which can be used for imitation learning of a policy. They demonstrate that the policies learned are able to go to different corners of different sized gridworlds.

**Strengths:**

I find there to be very few strengths in this paper to highlight, as I believe it falls far clear of the mark for a paper of appropriate quality for a conference of ICLR's standard. However, below, I raise a small number of positives.

- There are few typos or grammatical mistakes in the paper.
- The paper has seemingly good coverage of AI safety literature, though I believe this has little relevance to the paper itself.
- The paper is structured into easily followable sections.
- The authors make good use of figures to demonstrate their experiments.

**Weaknesses:**

As I stated earlier, this paper is not of a reasonable standard for admittance to ICLR 2025.

To summarise my interpretation of this paper, the writing is very hard to parse, the experiments are quite unclear, the results section is limited and the method is complicated and lacking in any kind of analysis or ablation. There is little grounding in relevant methodological literature, and in particular no comparison to prior work in goal-conditioned reinforcement learning; instead, there is a large focus on AI safety literature which, while important in general, has no relevance to the work at hand. Below, I raise a large number of specific weaknesses in order of presentation which I believe demonstrate its insufficient standard. As a general point, I would say that the focus on this work should be on attaining good policies as the links to AI safety feels forced and tenuous at best. I will highlight specific examples in my list below, but would say that a narrative shift would be of significant benefit to this paper.

# Motivation
- Linking RLHF to this system seems questionable (line 41). Firstly, it is quite difficulty to see how this method can be applied beyond gridworlds, though I am sure there are ways to do that. Secondly, RLHF uses a reward function based in human preferences; arguing that this 'may not always align with human values' seems counter to the entire basis of RLHF. For similar reasons, it is hard to argue that RLHF can lead to the loopholes discussed here.
- Introducing the paperclip maximiser scenario is a bombastic comparison when this paper is concerned with gridworlds only. Adding in a link to a game makes the paper seem unserious and again has no relevance to the work at hand.

# Method
- I will highlight all examples that stuck out to me, but there are *many* introduced components of this method and, while some are justified qualitatively, there is no example of any quantitative justification or ablation. It is impossible to see the importance of any part of this method.
- Discretising the state space is described as a key component (line 124) but doing this in a latent space is not ablated.
- Line 143, which stresses determinism by minimising entropy, is not ablated and seems quite hacky (eg the annealing schedule used).
- I am completely unclear why the deterministic decoder is written probabilistically?
- I am unsure how a deterministic decoder can ensure a bijective relationship. Surely you can still have a many-to-one relationship?
- If you already have a policy from Dijkstra's algorithm (ie you know the shortest route and actions to take it), why would you then imitation learn them? Entropy is included to account for sub-optimality but, again, Dijkstra's algorithm should yield an optimal route.
- The section on high-level behaviour seems out of place.

# Experiments
- It is not clear how the policy is trained for different goals. Is the graph recomputed anytime there is a new goal - this is exceptionally compute inefficient compared to goal-conditioned techniques.
- Related to the above, there is ***no*** baseline comparisons. There should be comparison with goal-conditioned reinforcement learning methods. [1] gives a brief overview of some relevant techniques.
- I really don't understand the high-level algorithms section. What is PPO doing in this context? Is it just drastically overfit to the situation where there is no blocking, experiencing policy collapse? It is not a surprise that this would fail.
- There are no error bars anywhere in this paper.
- It is not clear that this method would scale at all beyond a gridworld, or even to marginally larger gridworlds. Performance is already degraded at 25x25.
- There are no hyperparameters anywhere in the paper, making none of the experiments reproducible.
- It is unsurprising that PPO and DQN are outperformed by the proposed methods as they are designed *specifically* for the given task. This is especially true for HL-Simon, since that is designed to pick the policies in the order that their goals appear.
- There are no presented results for Simon besides qualitative statements that HL-Simon is the best.

# Related work
- There is no discussion of a number of key works under the field of 'goal-conditioned RL' which I think it would be important to highlight.

# Limitations

- There is a pretty glaring limitation of this work in that it only considers gridworlds rather than any more complicated environments. Thus, the results are very limited, in addition to the lack of comparative baselines.
- I feel an additional limitation is that, in my opinion, there is a very tenuous link between their motivation (AI safety) and their method.

# Broader Impact

- The broader impact statement in the appendix is, again, detached from the work. It is hard to suggest that this paper has implications in healthcare and education given its extremely limited setting.


Minor points to highlight:
- In the abstract (line 13), it is jarring to have the second sentence start 'E.g.,' and not read as a full sentence
- Line 80: saying 'We realize that rewards *also* have their downsides' doesn't make sense as the rest of the paragraph is based on the negatives of reward.
- In equation 1, the notation feels a bit odd and should be defined in text. Is $s_t$ the discretised predicted state? Not adding any accent makes it seem like this is a ground truth.
- Line 133 (below) is unclear.
>The discretization leads to partially overlapping simulation trajectories, that enable us to construct robust graphs for planning
-Referencing in the related work is improperly formatted, as papers are often referenced at the start and end of each sentence unnecessarily.

[1] Goal-Conditioned Reinforcement Learning: Problems and Solutions

**Questions:**

- How do you see this method being applied to situations in which distance (ie cost to go) is not a valid metric?
- Defining a goal and running the computation this way is impractical for many tasks. How do you see this method scaling beyond gridworlds to, say, a robotic dexterity task?
- For discretising the state space, does the fuzzy projection area not also just lead to large state changes (due to discretisation) from falling on either side of the boundary?
- What tuning was done to decide that the world model had 3 layers of 2048 hidden size? This seems exceptionally large given the limited number of state-action pairs, and likely converges to some tabular setting?
- Where does all the data for the world model come from? How much data is needed?
- What informed the decision to use an unconventional non-linearity (SiLU)?
- The process seems very inefficient. How much computation is needed?

---

### Official Review · Reviewer_HWuy · 2024-11-01

**Soundness:** 1
**Presentation:** 3
**Contribution:** 2
**Rating:** 3
**Confidence:** 4

**Summary:**

This paper presents a novel approach, RFPO, aimed at eliminating reward signals in policy learning while establishing a framework for developing high-level algorithms with human oversight. RFPO utilizes a backward world model and graph search techniques to condition policies on desired states, thereby mitigating reward hacking and unintended behaviors linked to reward signals. Experiments conducted in maze environments demonstrate that RFPO can effectively reach arbitrarily selected goals and adhere to human-designed algorithms.

**Strengths:**

1. This paper is well-motivated and easy to read.
2. The approach presented in the paper is simple and intuitively sound.

**Weaknesses:**

1. Availability. This method employs a Shortest Path Estimator (SPE) to determine the shortest paths from all nodes to the goal and conducts experiments in a maze environment (15 x 15, 20 x 20, 25 x 25). However, this environment may be too simplistic to provide substantial empirical evidence. I wonder whether this method can be scaled to tasks with larger state spaces.
2. Experiments. Additional training details should be provided, such as the number of training steps, hyperparameters used, and whether the maze environment was randomly generated during the training process.
3. Clarity. The implementation of high-level algorithms by human design is best placed in the ‘Method’ section. And their specific implementation needs to be further elucidated.

**Questions:**

1. It would be great if the author could address the major weaknesses I outlined above.
2. Can this method be evaluated in the existing benchmarks? Solely conducting experiments on self-designed environments probably not convincing enough.
3. What is the computational complexity of this method? Does the planning time consuming of the method increase exponentially with the state space to the point of unavailability?

---

### Official Review · Reviewer_C81h · 2024-11-04

**Soundness:** 2
**Presentation:** 3
**Contribution:** 1
**Rating:** 3
**Confidence:** 4

**Summary:**

Given that policy optimization in RL is susceptible to reward hacking, this paper proposes a novel policy optimization (RFPO) method based on imitation learning instead. Here, they leverage backward world models and discretized state spaces for planning in an MDP---planning using a shortest path estimator (SPE) like Dijkstra. The approach constructs trajectories in a backward manner from goal states, building a directed graph to optimize policies that can navigate toward predefined goals (i.e. the generated trajectories are used for imitation learning). The framework includes an encoder-decoder model that maps continuous states into discrete latent representations (similar to Dreamer v2), creating an actionable space for backward simulations. The authors demonstrate the method's effectiveness in maze environments, also introducing high-level algorithms that utilize RFPO to achieve human-aligned behaviors in a simple maze task (with a distribution shift between training and testing).

**Strengths:**

- **Originality**: The approach of using backward world models to derive policies in a reward-free setting is a creative departure from traditional imitation and reinforcement learning. By focusing solely on goal states rather than rewards, RFPO aligns well with goal-directed tasks.
- **Quality**: Methodology is sound, utilizing discretization to ensure planning is possible without requiring precise predictions at each time step. The structured graph-building method helps in identifying goal-reaching trajectories.
- **Clarity**: The paper provides a clear explanation of the backward planning process, including how graphs are built, SPE usage, and policy optimization through imitation learning. The visualization of maze environments in Figure 3 effectively demonstrates the policy's goal-reaching performance.
- **Significance**: This work advances imitation learning, addressing limitations in the generation of trajectories to learn on. The inclusion of high-level human-designed behaviors illustrates a good contribution towards controllable and interpretable policy execution.

**Weaknesses:**

- **Motivation and Setting**:
  - The authors motivate the problem setting using the reward hacking problem in RL. But they then only focus on goal-conditioned tasks and policies, instead of general tasks with more complex objectives (E.g. robot control and Atari games) that would normally require dense rewards in current RL methods. In goal-conditioned RL, one can simply use sparse rewards (e.g non-zero rewards only when reaching the goal) to avoid reward hacking from dense rewards. Can the authors clarify how their approach differs from or improves upon sparse reward goal-conditioned RL, particularly in terms of avoiding reward hacking?
  - In goal conditioned RL with sparse rewards, the objective of the agent is to both maximize the probability of reaching goals and also to do it in the shortest path. This paper only focusses on maximizing the probability of reaching goals, further reducing its applicability.
  - The first paragraph of Section 2 suggests that the dynamics and policy can be stochastic but this doesn't seem to align with the proposed approach (graph model and planning) and the conducted experiments. Can the authors explain how their method handles stochastic dynamics and policies if they are indeed supported? If stochasticity is supported, can the authors please explain why they chose to use deterministic models in their experiments?

  - Finally, the proposed approach is in imitation learning and doesn't seem to be related to RL. Can the authors clarify the relationship between their method and RL, or discuss how their approach compares to or complements existing RL methods?

- **Scalability Limitations**: The discretization method might not scale well to high-dimensional or complex continuous environments, where state overlaps and inaccuracies in backward planning could impede performance.

- **Experimental Scope**:
  - The environments are restricted to mazes, which, while effective for demonstrating goal-reaching, limit generalization claims.
  - Furthermore, these are extremely simple environments where even a random policy is optimal according to the problem definition (i.e it maximizes the probability of reaching a goal).
  - Evaluations in high dimensional continuous domains (such as robotic tasks), even with discrete actions, would better establish the model's utility.
  - Finally, the DQN and PPO baselines make very little sense. At the very least, the authors should have used similar high-level strategies for them. E.g Similar to HL-RND by adding random actions in test time to improve exploration, or Similar to HL-ADV by using an HRL method like option-critic. Additionally, DQN and PPO are Model-free. The authors should have used model-based baselines like Dreamer v2/v3, since their approach is also model-based.

- **Dependence on Graph Robustness**: The model heavily relies on the shortest path estimator (SPE) to curate goal-reaching paths. If the graph contains erroneous or noisy edges, the reliance on SPE may impact policy performance, especially in stochastic or partially observable environments.

- **Clarity on Latent Space Mapping**: While the encoder-decoder is theoretically justified, the model's actual performance may degrade if there is substantial variance in state encoding, especially under changes in the environment or extended exploration demands.

**Questions:**

Please refer to the weaknesses above. In addition,
- Can the authors include the performance of a random policy in the results, and the distance taken to reach the goal for all algorithms? If a random policy is optimal in these environments, can the authors explain why their method is necessary?
- Shouldn't the dynamics and policy in the first paragraph of Section 2 be assumed to be deterministic for the proposed approach?
- How would RFPO perform if state transitions were unpredictable or dynamic, such as in stochastic environments? Could backward planning still provide effective trajectories in such cases?
- Does the proposed RFPO method extend to environments with continuous actions, or are there inherent limitations that restrict it to discrete action spaces?
- Is there a critical threshold in the state space discretization where the model's performance drops significantly due to state overlap, and if so, how is this managed?

---

### Note · Authors · 2024-11-25

I have read and agree with the venue's withdrawal policy on behalf of myself and my co-authors.